# Immunotherapy for Metastatic Non-Small Cell Lung Cancer: Therapeutic Advances and Biomarkers

**Marco Russano** [1,*]**, Giulia La Cava** [1]**, Alessio Cortellini** [1]**, Fabrizio Citarella** [1]**, Alessandro Galletti** [2]**, Giuseppina Rita Di Fazio** [1]**, Valentina Santo** [1]**, Leonardo Brunetti** [1]**, Alessia Vendittelli** [1]**, Iacopo Fioroni** [1]**, Francesco Pantano** [1]**, Giuseppe Tonini** [1] **and Bruno Vincenzi** [1]

[1] Department of Medical Oncology, Campus Bio-Medico University of Rome, Via Álvaro del Portillo, 21, 00128 Rome, Italy
[2] Division of Medical Oncology, San Camillo Forlanini Hospital, 00152 Roma, Italy
[*] Correspondence: m.russano@policlinicocampus.it; Tel.: +39-06225411252

**Abstract:** Immunotherapy has revolutionized the treatment paradigm of non-small cell lung cancer and improved patients' prognosis. Immune checkpoint inhibitors have quickly become standard frontline treatment for metastatic non-oncogene addicted disease, either as a single agent or in combination strategies. However, only a few patients have long-term benefits, and most of them do not respond or develop progressive disease during treatment. Thus, the identification of reliable predictive and prognostic biomarkers remains crucial for patient selection and guiding therapeutic choices. In this review, we provide an overview of the current strategies, highlighting the main clinical challenges and novel potential biomarkers.

**Keywords:** immunotherapy; non-small cell lung cancer; immune checkpoint inhibitors; PD-1/PD-L1 antibodies; combination strategies; biomarkers; tissue biomarkers; liquid biopsy

## 1. Introduction

Lung cancer is the leading cause of cancer-related deaths worldwide [1]. With 2.2 million new cancer cases and 1.8 million deaths, it represents approximately one in 10 (11.4%) cancers diagnosed and one in 5 (18.0%) deaths. The incidence is higher in industrialized and transitioned countries, reflecting the high burden of risk factors, including mainly tobacco smoking but also other inhalable agents such as asbestos and air pollutants. However, this pattern may well change as the tobacco epidemic grows in the low-income and middle-income countries (LMICs) [2]. The poor prognosis is due to biological aggression, the high tendency to metastasize and the long asymptomatic latency. For these reasons, most patients have metastatic disease at the time of diagnosis and less than 20% of them are alive at 5 years [3].

The treatment depends on the tumor histology and molecular profile. Lung tumors are categorized into two major histological groups: small cell lung cancer (SCLC) and non-small cell lung cancer (NSCLC). NSCLC accounts for 80 to 85 percent of cases, and includes two histologic subtypes: squamous cell carcinoma and non-squamous carcinoma, mainly adenocarcinoma [4]. A subset of NSCLC hosts a single-driver anomaly that dictates cell growth, survival, and metastasization. This phenomenon, called oncogene-addiction, more often occurs in adenocarcinoma histology, in non-smokers, Asian ethnicity, young, and female patients [5,6]. Mutations on EGFR, ALK, ROS-1, BRAF and novel emerging drivers, are therapeutic targets for Tyrosine-kinase Inhibitors (TKI), and several studies have shown greater efficacy and a better safety profile for these drugs than chemotherapy (ChT). As a result, TKIs have become the standard of care for oncogene-addicted NSCLC [7].

However, most patients have non-oncogene-addicted disease. They cannot benefit from targeted therapies and chemotherapy shows limited effect on survival rates. Certainly, the advent of anticancer immunotherapy has dramatically changed the treatment landscape

and improved patients' prognosis. Although it is still difficult to accurately estimate the overall survival benefit, some studies have reported increased survival rates 5 times higher than those achieved with chemotherapy [8,9]. Nevertheless, only a minority of patients achieve a long-term durable response; most of them do not respond or develop progressive disease during treatment. Then, the identification of reliable predictive and prognostic biomarkers remains the crucial item for patient selection and guiding therapeutic choices.

## 2. Immune Checkpoints

Modern immuno-oncology (IO), consists of different strategies such as vaccines, CAR-T and immunomodulators, aimed at mobilizing and activating the immune system to recognize and destroy cancer cells [10–13]. In current clinical practice, the main therapeutic approach is the use of monoclonal antibodies that act on specific key modulators of T-cell immune response. These molecules, also known as immune checkpoint (IC), are present on T cells, antigen-presenting cells (APCs) and tumor cells. Their interaction activates either inhibitory or activating immune signaling pathways [14–16]. Most relevant checkpoints are the inhibitory pathways consisting of cytotoxic T lymphocyte-associated molecule-4 (CTLA-4), programmed cell death receptor-1 (PD-1), and programmed cell death ligand-1 (PD-L1).

The first immune checkpoint inhibitor (ICI) to enter clinical practice was Ipilimumab, an anti-CTLA-4 antibody, which received FDA approval as monotherapy for late-stage melanoma in 2011 [17,18]. Later on, ICIs dominated clinical research and the therapeutic scenario in many malignancies. For NSCLC patients, the greatest successes have been achieved with inhibitors of the PD-1/PD-L1 axis. Anti-PD-1/PD-L1 antibodies offered significant advantages over chemotherapy, including a favorable safety profile, increased antitumor activity, induction of long-lasting responses, and improved survival. Several trials have evaluated their efficacy as a monotherapy, as well as in combination with other agents, and led to upset the algorithms of treatment [19].

New immune checkpoint targets are under study, like lymphocyte activation gene-3 (LAG-3), T-cell immunoglobulin and mucin-domain containing-3 (TIM-3), T-cell immunoglobulin and ITIM domain (TIGIT), and V-domain Ig suppressor of T-cell activation (VISTA) [20]. The lymphocyte activation gene-3 (LAG-3), a cell-surface protein expressed on immune cells, which negatively regulates T-cell proliferation, was proven to be an attractive target. Relatlimab, a LAG-3 antibody, demonstrated survival benefits in previously untreated melanoma when combined with nivolumab, and is now being evaluated in NSCLC patients [21,22]. The V-domain Ig suppressor of T-cell activation (VISTA), as well known as PD-1 homologous (PD-1H), is a co-inhibitory molecule, mostly expressed on neutrophils, monocytes, macrophages, APCs and to a lower extent, on naive CD4+ and CD8+ T cells, but not on tumor cells; and regulates chemotactic activity of T cells, being able to suppress the immune response [23,24]. Therefore, targeting PD-H1 and silencing their action would allow the chemotactic migration of T cells, promoting tumor infiltration and enhancing the ability of T cells to restore their antitumoral activity. T-cell immunoglobulin and mucin-domain containing-3 (TIM-3) was found to have an important role as a negative regulatory immune checkpoint present on T cells, DCs, B cells and NK cells, by mediating T-cell exhaustion [25]. Different preclinical studies have shown how TIM-3 inhibitors have a similar efficacy as that of PD-1 inhibitors, enabling T-cell response enhancement [26]. Another emerging immune checkpoint target is the T-cell immunoglobulin and ITIM domain (TIGIT), a co-inhibitory receptor that competes with the costimulatory function of CD226, playing an important role in T-cell activation and maturation [27]. The synergistic block of TIGIT and PD-L1 appears able to reactivate T cells and enhance NK cells' actions. In a phase II trial, adding Tiragolumab (anti-TIGIT) to atezolizumab (anti PD-L1 antibody) has already shown improved objective responsive rates and progression-free survival in treatment-naïve metastatic NSCLC patients [28].

### 2.1. Anti PD-1/PD-L1 Antibodies

Blockade of the PD-1/PD-1 axis represents the mainstay of the current immunotherapy of NSCLC. PD-1 is a membrane receptor found on the T-lymphocyte surface during the effector phase, natural killer (NK), B-lymphocytes, and APC cells and requires binding with its ligands PD-L1 and PD-L2, expressed in tumor cells. Their interaction plays a key role as an inhibitor of both adaptive and innate immune responses. It prevents the proliferation, differentiation and activation of T cells, hinders cytokine production and can inhibit signaling through the B-cell receptor [29,30].

Nivolumab is a fully human IgG4 monoclonal antibody against PD-1. In two large, international phase III trials, CheckMate 017 [31] and CheckMate 057 [32], Nivolumab showed improved survival over Docetaxel after a first-line platinum-based ChT in squamous and non-squamous NSCLC, respectively. Median overall survival (mOS) was more than two months longer in the Nivolumab groups: mOS was 9.2 months versus 6 months in squamous NSCLC and 12.2 months versus 9.4 in non-squamous NSCLC. Based on these results, Nivolumab was the first ICI to receive FDA approval in second-line treatment for advanced disease [33]. Subsequently, two other ICIs confirmed the superiority of immunotherapy over Docetaxel in pretreated patients. Atezolizumab is a monoclonal antibody that binds to PD-L1. In the OAK trial [34], patients previously treated with one to two ChT regimens were randomized to receive Atezolizumab or Docetaxel. OS was significantly higher in the Atezolizumab group compared with the Docetaxel group (13.8 months [95% CI, 11.8–15.7] vs. 9.6 months [8.6–11.2]). The study showed survival benefits regardless of PD-L1 expression or histology. Thus, like Nivolumab, Atezolizumab was approved as a second-line therapy regardless of PD-L1 status.

Pembrolizumab, a highly selective anti-PD-1 humanized monoclonal antibody, is the first ICI to show differentiated efficacy based on PD-L1 expression. The Keynote 010 trial [35] was conducted on patients with previously treated NSCLC and PD-L1 expression on at least 1% of tumor cells (TC). Pembrolizumab showed to improve OS compared with Docetaxel in the total population, but with a huge benefit, especially in patients with at least 50% of TC expressing PD-L1. Consequently, the Keynote 024 [36] tested Pembrolizumab as a first-line treatment in patients with PD-L1 expression on ≥50% of TCs. Immunotherapy was associated with significantly longer progression-free survival (PFS) and OS, and a better safety profile than platinum-based doublet ChT. New 5-year data showed roughly double survival rates: median OS was 26.3 months (95% CI, 18.3 to 40.4) for Pembrolizumab and 13.4 months (9.4–18.3) for chemotherapy (hazard ratio, 0.62; 95% CI, 0.48 to 0.81) [37]. These results established Pembrolizumab as a standard first-line therapy in patients with high PD-L1 expression.

Recently, another anti-PD-1 antibody, Cemiplimab, showed survival benefits in the first-line treatment of mNSCLC with PD-L1 of at least 50%. At median follow-up of 37.1 months, mOS was 26.1 m for the Cemiplimab group vs. 13.3 m for patients treated with chemotherapy (hazard ratio [HR] 0.57; $p = 0.0001$) [38]. Similarly, Atezolizumab significantly prolonged overall survival vs. chemotherapy in patients with high PD-L1 expression (PD-L1 expression in at least 50% of tumor cells or at least 10% of tumor-infiltrating immune cells). Median overall survival in this group was 20.2 months vs. 13.1 months (stratified hazard ratio for death, 0.59; 95% confidence interval [CI], 0.40 to 0.89; $p = 0.01$) [39].

All of these ICIs were compared with chemotherapy and received regulatory approval. In the absence of head-to-head trials between antibodies, it is difficult to determine which is the best treatment. Therefore, clinicians have more immunotherapeutic options in both first- and second-line treatment for stage IV NSCLC.

To date, only an anti-PD-L1 antibody, Durvalumab, has been approved for non-resectable stage III NSCLC. In the PACIFIC trial, 713 patients received Durvalumab (n = 476) or a placebo (n = 237) as consolidation therapy following chemoradiation. The study found that Durvalumab significantly prolonged overall survival, as compared with the placebo (stratified hazard ratio for death, 0.68; 99.73% CI, 0.47 to 0.997; $p = 0.0025$) [40]. Updated

exploratory analyses demonstrated a sustained survival benefit: estimated 4-year OS rates were 49.6% versus 36.3% for Durvalumab versus a placebo [41].

Advances in the treatment of advanced disease anticipated trials of immunotherapy in early-stage NSCLC. Ongoing studies are evaluating the role of PD-1/PD-1 blockade as an adjuvant or neoadjuvant treatment [42]. The superiority of ICI over conventional treatments has accumulated evidence in non-oncogene addicted disease. Efficacy data in tumors harboring oncogenic drivers are limited. In these cases, target therapies remain the standard of care, and immunotherapy should only be considered at the failure of other treatment options [43]. Conversely, all patients with non-oncogene addicted disease could benefit from ICI treatment, with the exception of those having contraindication (active and serious autoimmune disorders). However, despite the advances, only a minority of patients respond to single-agent ICI. To improve efficacy, the combination of chemotherapy and immunotherapy was expected to increase the responders' rates and further prolong survival [44]. Combining ICIs with chemotherapy exploits the additive effect but also has the potential for synergy through several mechanisms. In particular, chemotherapy could elicit immune response against cancer by increasing antigen presentation to T cells, inducing immunogenicity and relieving tumor-induced immunosuppression [45–47].

### 2.2. Combination Strategies

Several randomized clinical trials (RCT) have demonstrated an OS benefit for the addition of anti-PD-1/PD-L1 antibodies to platinum-based chemotherapy, providing new frontline treatment strategies for metastatic NSCLC.

The KEYNOTE-189 study [48] is a randomized double-blind phase 3 trial in non-squamous patients. Pembrolizumab in combination with platinum and Pemetrexed improved response rates (RRs) from 18.9% to 47.6% and increased PFS, reaching a median value of 8.8 months in the Pembrolizumab-combination group versus 4.9 months in patients treated with chemotherapy alone. The 12-month overall survival was 69.2% versus 49.4% (hazard ratio for death, 0.49; 95% CI, 0.38 to 0.64; *p* < 0.001) [41]. The 5-y OS rates were 19.4% in the Pembrolizumab arm versus 11.3% in the chemotherapy-alone. The OS benefit was irrespective of PD-L1 expression, with the greatest improvement in patients with a PD-L1 TPS of at least 50% [49].

Superiority of the combination strategy was also shown in squamous NSCLC. In the KEYNOTE-407 trial [50], the addition of Pembrolizumab to Carboplatin and Paclitaxel or Nab-paclitaxel improved median OS from 11.3 to 15.9 months after a median follow-up of 7.8 months (HR for death 0.64). survival and response rates remained improved at longer follow-up. 5-y OS rates were 18.4% and 9.7% in Pembrolizumab/chemotherapy and placebo/chemotherapy, respectively [51]. Also in this trial, the survival benefit was consistent regardless of the level of PD-L1 expression.

Combinations of platinum-based ChT doublet and anti-PD-L1 antibodies, including both Atezolizumab and Durvalumab, have been troubled in development. In two randomized phase 3 trials, Atezolizumab plus chemotherapy for first-line treatment of metastatic NSCLC showed to improve PFS but did not meet the OS endpoint [52]. Instead, the IMpower130 trial, a multicenter, randomized, open-label, phase 3 trial, showed both PFS and OS superiority of Atezolizumab in combination with Carboplatin plus Nab-paclitaxel chemotherapy compared with chemotherapy alone as first-line treatment for metastatic non-squamous NSCLC [53].

EMPOWER-Lung 3, a double-blind phase 3 trial compared Cemiplimab plus platinum-doublet chemotherapy versus chemotherapy-alone as first-line treatment for mNSCLC, irrespective of PD-L1 expression or histology. The trial was stopped early because it met preset OS efficacy criteria: median OS was 21.9 months in the Cemiplimab plus chemotherapy group versus 13 months in the group treated with chemotherapy alone (hazard ratio (HR) = 0.71; 95% CI, 0.53–0.93; *p* = 0.014) [54].

Similarly, Sugemalimab (anti-PD-L1 antibody) was combined with platinum-based chemotherapy and compared with chemotherapy alone for naïve mNSCLC. The median

OS was 25.4 vs. 16.9, respectively (HR = 0.65 [95%CI, 0.50–0.84], *p* = 0.0008), and 2-y OS rate was 51.7% vs. 35.6%. Survival benefit was observed in both squamous and non-squamous histology [55].

In all these trials, the OS benefit was seen across all relevant patient subgroups, including those with low or negative PD-L1 expression. Furthermore, although immunotherapy adds IrAEs to chemo-related toxicities, combination treatments have shown acceptable tolerance profiles. Then, anti-PD-1/PD-L1 antibodies plus platinum-based doublet ChT became a standard first-line treatment for metastatic NSCLC. However, combination regimes were always compared to chemotherapy alone. No direct comparison has been made with single-agent ICI in patients with PD-L1 TPS of ≥50%. So, anti-PD-1/PD-L1 monotherapy remains an appropriate option in this setting. Indeed, for the same reason, in countries, including Italy, the addition of Pembrolizumab to platinum-based chemotherapy received regulatory approval only in patients with PD-L1 expression less than 50%, making the use of a single-agent antibody the only one immunotherapeutic chance for patients with higher PD-L1 expression.

Recent advances derive from ICI combinations, especially anti-PD-1 and anti-CTL-4 antibodies. It has been found that the simultaneous CTLA-4 and PD-1/PD-L1 blockade results in a synergistic and enhanced immune response against cancer: inhibition of CTLA-4 mainly acts on the lymph nodes and can upregulate PD-1 expression favoring the activation of T cells; therefore, it exerts a priming effect for the PD-1 blockade, which improves the function of effector T cells in the tumor microenvironment [56]. The efficacy of dual immunotherapy has already been explored in large RCTs. When compared to chemotherapy, it appears to provide controversial results, showing survival benefit especially in some subgroups of patients. When combined with chemotherapy, it clearly and significantly improved overall survival.

Checkmate 568, an open-label phase II trial, successfully tested the efficacy and safety of Nivolumab plus low-dose Ipilimumab as a first-line treatment of advanced/metastatic NSCLC [57]. Survival benefit over chemotherapy was confirmed in the CheckMate 227 trial, and was independent of the PD-L1 expression level [58].

A recent phase 3 trial (Checkmate 9LA) explored the effect of Nivolumab plus Ipilimumab combined with two cycles of platinum-based chemotherapy in the first-line setting for mNSCLC. Immuno-chemotherapy provided a significant improvement in overall survival and had a favorable safety profile. At a median follow-up of 13.2 months, median OS was 15.6 months in patients treated with the combined strategy group vs. 10.9 months in patients treated with chemotherapy alone (HR = 0.66, 95% CI = 0.55–0.80). Overall survival benefit in the Nivolumab/Ipilimumab plus chemotherapy group was observed in both squamous and non-squamous histologies and across all PD-L1 expression levels [59]. These new data broaden first-line treatment options for metastatic NSCLC [60].

Another dual immunotherapy consisting of anti-PD-L1 (Durvalumab) and anti-CTLA-4 (Tremelimumab) antibodies was investigated in two phase 3 randomized clinical trials. In the Mystic trial, patients with untreated mNSCLC were randomized to receive treatment with Durvalumab, Durvalumab plus Tremelimumab, or chemotherapy. Despite treatment with Durvalumab resulting in a reduced risk of death in patients with PDL-1 expression on at least 25% of tumor cells, the study did not meet its primary end points of improved OS with immunotherapeutic strategies vs. chemotherapy [61]. The POSEIDON trial showed that Durvalumab plus chemotherapy significantly improved PFS while a positive trend for OS did not reach statistical significance. Instead, Durvalumab combined with Tremelimumab and chemotherapy versus chemotherapy alone improved both PFS and OS (median OS of 14 months versus 11.7 months) [62]. Most recently, a PD-1/CTLA-4 bispecific antibody (MEDI5752) in combination with platinum-based chemotherapy was compared to Pembrolizumab plus chemotherapy in treatment-naïve non-small cell lung cancer. Interim analysis showed improved PFS e OS, especially in the PD-L1 < 1% subgroup [63].

These results highlight the usefulness of immunotherapeutic approaches in the first-line treatment for mNSCLC patients. To maximize the potential effects of immunotherapy,

combination with angiogenic drugs has also been studied. As a matter of fact, interaction between immune cells and newly generated tumor vessels represents an important obstacle that can decrease efficacy of ICIs treatment through many different mechanisms. A malformed network of newly generated tumor microvessels serves as a physical barrier for T cells traveling in the bloodstream, preventing them from infiltrating the tumor and exerting their action [64]. Furthermore, angiogenetic signals may directly elicit immunosuppression: an hypoxic state in the tumor microenvironment stimulates production of vascular-endothelial growth factor (VEGF) and newly formed vessels express different immunosuppressive molecules such as PD-L1 [65]. Hence, the combination of immunotherapy and antiangiogenesis is a promising strategy for cancer treatment based on a strong biological rationale for the synergistic effect. However, the addition of angiogenic drugs to ICIs has achieved unsatisfactory effects in clinical trials, remaining an attractive area of research [66,67].

Immunotherapeutic strategies that showed to improve survival in phase III trials are listed in Table 1.

**Table 1.** First-line immunotherapy for mNSCLC: phase 3 trials showing OS benefit from ICIs.

| TRIAL | DETAILS | HISTOLOGY | PD-L1 | mOS (Months) |
|---|---|---|---|---|
| KEYNOTE 024 | Pembrolizumab vs. Platinum-based ChT | NSCLC | PD-L1 $\geq$ 50% | 26.3 vs. 13.4 |
| IMPOWER110 | Atezolizumab vs. Platinum-based ChT | NSCLC | PD-L1 $\geq$ 50% | 20.2 vs. 13.1 |
| EMPOWER-LUNG1 | Cemiplimab vs. Platinum-based ChT | NSCLC | PD-L1 $\geq$ 50% | 26.1 vs. 13.3 |
| KEYNOTE 189 | Pembrolizumab + Platinum-pemetrexed vs. Placebo + Platinum-pemetrexed | NS-NSCLC | All comers | 22.0 vs. 10.6 |
| KEYNOTE 407 | Pembrolizumab + CBCDA and paclitaxel vs. Placebo + CBCDA and paclitaxel | Squamous NSCLC | All comers | 17.1 vs. 11.6 |
| IMPOWER130 | Atezolizumab + CBCDA plus nab-paclitaxel vs. CBCDA plus nab-paclitaxel | NS-NSCLC | All comers | 18.6 vs. 13.9 |
| EMPOWER-LUNG3 | Cemiplimab + Platinum-based ChT vs. Cht alone | NSCLC | All comers | 21.9 vs. 13 |
| GEMSTONE 302 | Sugemalimab + ChT vs. Platinum-doublet ChT | NSCLC | All comers | 25.4 vs. 16.9 |
| POSEIDON | Durvalumab + Tremelimumab + ChT vs. Platinum-doublet ChT | NSCLC | All comers | 14.0 vs. 11.7 |
| CHECKMATE 227 | Nivolumab + Ipilimumab vs. Platinum doublet ChT | NSCLC | PD-L1 $\geq$ 1% | 17.1 vs. 13.9 |
| CHECKMATE 9LA | Nivolumab + Ipilimumab and Platinum doublet ChT vs. Platinum doublet ChT | NSCLC | All comers | 15.9 vs. 10.9 |

### 2.3. Treatment Paradigm and Clinical Challenges

ICIs have quickly revolutionized the treatment paradigm for stage IV NSCLC. Single agents and combination strategies have both demonstrated great clinical activity and manageable side effects. The survival improvements shown in several phase 3 RCT have led to multiple regulatory approvals. As a result, clinicians have more immunotherapeutic options in treating metastatic disease.

In the current algorithms, the distinction between non-oncogene-addicted and oncogene-addicted disease remains a cornerstone. Target therapies are the unquestioned standard of care for patients with actionable mutations, and immunotherapy should only be considered upon failure of other treatment options. A potential ICI activity in oncogene-addicted NSCLC was documented in the phase 3 IMpower150 study: the addition of Atezolizumab

to Bevacizumab plus chemotherapy resulted in a survival improvement in NSCLC patients, including patients with EGFR mutations or ALK translocations [68].

Although the role of immunotherapy remains controversial in this setting, combination strategies with target therapies may increase antitumor activity and represent a new field of research. However, data on the efficacy of combining PD-1 inhibitors with Tyrosine-kinase Inhibitors (TKIs) are mainly obtained from subgroup analysis and revealed a limited effect [69]. To date, combinations with anti-PD-1/PD-L1 antibodies and anti-EGFR TKIs have failed to improve survival and showed a low safety profile [70,71]. Other combinations with anti-EGFR TKIs or ALK inhibitors are currently being investigated in ongoing phase I-II clinical trials [72]. Further research should focus on emerging oncogenic aberrations, including RET fusions, ROS and NTRK rearrangements, MET, BRAF and KRAS mutations, for which the role of immunotherapy is poorly explored. Furthermore, the advent of new genomic diagnostic tools, including next generation sequencing (NGS), has improved knowledge of cancer biology and implemented the acquisition of molecular targets [73,74]. By providing a comprehensive molecular profile, NGS also offers the opportunity to identify potential biomarkers for immunotherapy, both from tissue and blood samples [75,76]. Based on recent evidence, international guidelines already recommend the use of NGS to extend the molecular profile, especially in order to favor access to target therapies [77]. Instead, the predictive and prognostic role for patients treated with immunotherapy has not yet been clarified. Although it is not routinely used in clinical practice and only a few centers can benefit from it, in the near future, NGS could become a standard method as a molecular diagnostic tool for NSCLC.

Therefore, immunotherapy currently plays a marginal role in oncogene-addicted disease, but most patients do not have a molecular target and do not receive targeted therapies (non-oncogeneaddicted disease). In these cases, ICIs are standard of care for metastatic NSCLC, either as a single-agent treatment or in combination strategies. Monotherapies in pre-treated patients have been widely used in the past, but are now outmoded as ICIs are placed at the first-line setting. Combined therapy has shown to improve the effectiveness of immunotherapy, expanding the beneficiary population and overcoming drug resistance. Combinations of ICIs with platinum-based doublet chemotherapy represent appropriate therapeutic choices in both squamous and non-squamous histology, regardless of PD-L1 expression. They are superior to chemotherapy alone even in PD-L1 negative patients, but their efficacy is greater in PD-L1 positive patients. However, there is no RCT comparing the combination approach and anti-PD-1/PD-L1 single-agent in patients with PD-L1 expression $\geq$ 50% on TCs. As a result, ICI monotherapy remains a therapeutic option in this setting, especially in patients unfit for chemotherapy [78,79]. In this regard, most clinical studies have excluded frail patients. Consequently, there is little evidence of ICI efficacy in special populations, including ECOG PS 2 and elderly patients. In the real world, they have poor therapeutic chances, so it is necessary to strengthen the research [80,81]. Thankfully, the results of a study dedicated to patients ineligible for platinum-based chemotherapy were recently presented at the ESMO Congress 2022: the IPSOS trial showed improved OS in patients receiving first-line Atezolizumab rather than single-agent chemotherapy (mOS 10.3 vs. 9.2 months, HR 0.78, *p* = 0.028; 2y-OS 2 24.3% versus 12.4%) [82].

Although ICIs have become standard of care, some areas of unmet need remain. For instance, we should optimize the monitoring and measuring of the response to treatment since there may be atypical response patterns, such as pseudoprogression and hyperprogression, albeit rare [83]. Pseudoprogression (PsP) is defined as an initial progression of disease, followed by a decrease in tumor burden. Although the exact pathophysiological mechanism is still unclear, this phenomenon may be caused by the infiltration of immune cells causing an apparent increase in tumor size [84]. Furthermore, the antitumor response to immunotherapy treatment may occur later than to chemotherapy or targeted therapies. PSP has emerged as a novel tumor response pattern during immunotherapy, but it so difficult to distinguish from true progression, representing a challenge for clinicians. Thus, novel radiological immune-related response criteria to accurately evaluate the response

to ICIs have been proposed [85]. In particular, according to immune-RECIST (iRECIST) criteria, radiological progression of disease needs to be confirmed after a first evaluation, on the basis of observing either a further increase in size or in the number of new lesions [86]. However, PSP was first described in melanoma and appears much less common in NSCLC patients treated with ICI, with incidence rates ranging from 1 to 5% in clinical trials [87]. If PsP occurs in patients who benefit from immunotherapy, hyperprogression is instead the expression of true aggressive tumor behavior and a lack of response to treatment. Rather, hyperprogressive disease (HPD) is a pattern of rapid tumor progression and prompt clinical deterioration during ICI treatment [88]. The existence of this phenomenon had been proved but the definition of HPD is heterogenous and based on the different assessment approaches, such as tumor growth kinetics (TGK) and tumor growth rate (TGR) [89]. Consequently, there is no agreement on the incidence and methods for validation. The mechanisms causing HPD are yet to be elucidated, but since it is a specific pattern of ICI treatment, research is focusing on immunological pathways that could act as accelerators of tumor growth and metastasis [90].

We should also redefine the absolute contraindications to treatment since not all autoimmune disorders interfere with ICI. Moreover, the mechanisms underlying immune-related adverse events (IrAEs) remain unclear. Due to exuberant immune stimulation, ICI can cause inflammatory events that resemble autoimmune diseases, especially thyroiditis, skin disorders, colitis and pneumonitis. Any organ can potentially be affected, including kidney (nephritis), liver (hepatitis), musculoskeletal and nervous systems [91,92]. Serious events may cause treatment discontinuation and require management with high doses of steroids or immunosuppressive agents. Then, IrAE prediction, recognition and monitoring are crucial but are still matter of debate.

Last but not least, some patients do not respond to treatment, and only a minority have a durable response. Combination therapy approaches (chemotherapy and immunotherapy) were born to increase response rates to ICIs and improve survival, but the identification of a predictive biomarker is essential for distinguishing responders from non-responders [93]. The PD-L1 expression on tumor cells is the only biomarker validated and broadly applied in clinical practice, but it has some limitations indicating the need for better biomarkers to guide patient selection.

### 3. Immune Biomarkers

Predictive and prognostic biomarkers have been the main field of research in immuno-oncology for years. A large number of cells and particles participate in the tumor-immune interaction. Therefore, several molecules and parameters could correlate with clinical outcomes and predict response to ICI treatment. There are broad categories hosting potential biomarkers, including tumor microenvironment (TME), immune checkpoints, cancer neoantigens, tumor mutational burden, inflammation, and multiple tumor-associated components circulating in the bloodstream [94,95]. At present, only a few tissue-based biomarkers have been shown to predict the efficacy of ICIs in NSCLC patients, mainly including the PD-L1 expression and tumor mutation burden (TMB). Many other potential biomarkers are under investigation, both from tissue samples and liquid biopsies (Figure 1).

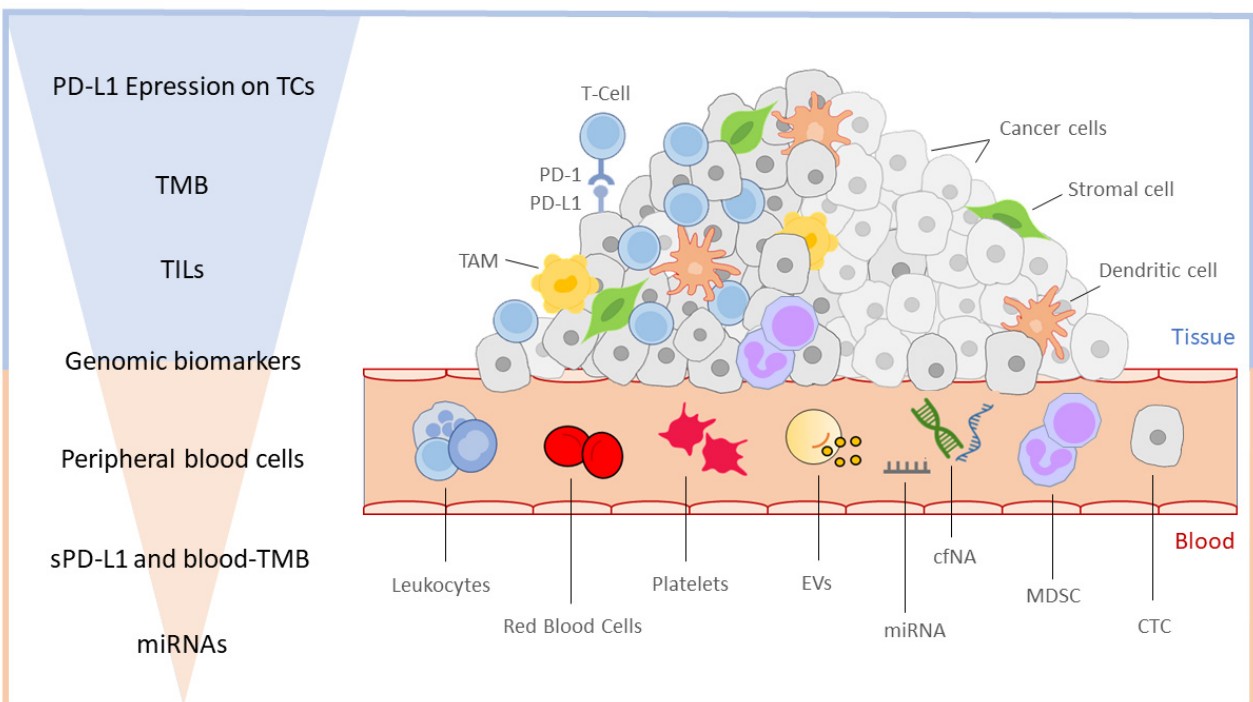

**Figure 1.** Tissue- and blood-derived biomarkers for ICIs in NSCLC. The figure shows biomarkers detectable using tissue and blood biopsies. On the left side, the biomarkers are listed in descending order, first the tissue and then the blood ones, according to the amount of evidence and robustness of the literature data. PD-L1 expression on tumor cells remains the main biomarker, currently the only one validated in clinical practice. Genomic biomarkers including MSI and other aberrations lie in the middle position because they are findable by both tissue and liquid biopsy. On the right side, the tumor microenvironment is represented. The cancer mass differs in a cold tumor and hot tumor, the latter characterized by a higher density of tumor-infiltrating lymphocytes (TILs) and other immune cells, such as tumor-associated macrophages (TAMs). Many cell populations from TME, including immune cells and circulating tumor cells (CTCs), flow into the bloodstream and can be analyzed through liquid biopsy. Promising data on biomarkers research comes from cell-free nucleic acids (cfNA), especially microRNAs (miRNA) derived from extracellular vesicles (EVs), especially exosomes.

### 3.1. Tissue Biomarkers

The main biomarker widely used in clinical practice is the PD-L1 expression on TCs or infiltrating immune cells, assessed by immunohistochemistry (IHC) staining [96]. Several prospective trials demonstrated a correlation with efficacy of anti-PD-1/PD-L1 treatment. The first strong evidence comes from the Keynote 010 and Keynote 024 trials, which led to the approval of Pembrolizumab in the second-line setting for PD-L1 positive patients and as a frontline treatment for patients with PD-L1 expression $\geq$ 50%, respectively [97]. Later, Atezolizumab and Cemiplimab also showed improved survival compared with platinum-based chemotherapy among NSCLC patients with high PD-L1 expression. Multiple exploratory analyses support these findings. Among these, in the PACIFIC trial, an exploratory post-hoc analysis requested by EMA revealed no survival benefit from Durvalumab after chemoradiotherapy for unresectable NSCLC in patients with PD-L1 expression < 1% [98]. In contrast, a proportion of PD-L1 negative tumors can also respond to anti-PD-1/PD-L1 treatments, including Nivolumab and Atezolizumab as single agents after platinum-chemotherapy failure [99,100]. Additionally, the CheckMate 227 trial reported OS benefit from Nivolumab plus Ipilimumab compared to chemotherapy in both PD-L1 positive and PD-L1 negative tumors [101]. In general, most evidence shows that PD-L1 expression may be a predictor of the efficacy of anti-PD-1/PD-L1 antibodies when used

as single-agent [102]. However, with the advent of combination strategies, its predictive value is significantly diminished. ICI plus platinum-based chemotherapy reported survival improvement regardless of PD-L1 expression. Although the efficacy increases with higher PD-L1 levels, it has also been proven in PD-L1 negative tumors [103].

Whether PD-L1 expression is the best biomarker for lung cancer immunotherapy is still a matter of debate as several controversies limit its reliability [104–106]. Studies have shown the heterogeneous nature of PD-L1 expression, which differs within tumors, may be inconsistent in sections of the same tumor sample, and can vary during treatment [107,108]. Finally, clinical trials used various cut-off points and detection methods resulting in the need for standardization [109].

In the last years, tumor mutational burden (TMB) has arisen as another biomarker for immunotherapy. It is defined as the total number of somatic mutations in a tumor sample. A high number of mutations results in a greater number of tumor neoantigens and increased activation of immune cells. Thus, TMB reflects neoantigen load and represents a surrogate for cancer immunogenicity [110,111]. The evidence for TMB as an immune-related biomarker in NSCLC mainly originates from retrospective or subgroup analysis of RCTs. The CheckMate 026 trial tested first-line Nivolumab in patients with a PD-L1 expression of 5% or more. Nivolumab did not show significantly longer PFS and OS than platinum-based chemotherapy. An exploratory analysis revealed that among patients with a high burden of tumor mutations (243 or more mutations), the response rate and PFS were higher in the Nivolumab group, but no OS differences were observed [112]. Similarly, in the CheckMate 227, high TMB ($\geq$10 mutations per megabase) was associated with longer PFS in patients receiving a combination of Nivolumab plus Ipilimumab compared with chemotherapy, but failed to predict OS [58].

Two recent systematic reviews and metanalysis have proven a clinical and survival benefit from ICIs in patients with high TMB compared with those with low TMB [113,114] but more prospective data are warranted. Additionally, no correlation with PD-L1 expression has been shown and other issues limit its utility and validation. With the advent of new molecular diagnostic tools such as NGS, TMB will be increasingly available in clinical practice. However, at present, there is no standardization across the testing platforms, and there is no consensus to define "high" TMB as a different cut-off has been used in clinical trials [115,116].

Mismatch repair deficiency (MMRD) and microsatellite instability (MSI) have been associated with an increased TMB and may be determinants of tumor immunogenicity [117]. Consequently, some clinical trials have studied their potential role as predictive biomarkers for ICIs. Recent evidence reported a high response rate and increased survival in several MSI-high and MMRD solid cancers, leading to the first tissue/site-agnostic approval by the FDA [118,119]. The KEYNOTE-177 trial found that first-line Pembrolizumab significantly improved PFS compared with chemotherapy for MSI-H–dMMR metastatic colorectal cancer [120]. Nevertheless, MMRD and MSI are rare in lung cancers and poorly studied as potential biomarkers for NSCLC ICIs [121].

Emerging biomarkers arise from the characterization of TME cell populations. TME is highly heterogeneous and dynamic, composed of immune cells, fibroblasts, and vascular and lymphatic vessels surrounding the tumor cells. Among these, tumor-infiltrating lymphocytes (TILs) have been correlated with ICI efficacy in several cancer types, including NSCLC [122,123]. High levels of TILs, particularly CD8 + T cells, reflect an increased immune response against cancer and characterize the inflamed phenotype, the so-called "hot" tumor. Categorizing tumors into "hot" and "cold" tumors through the identification of tissue biomarkers could help the selection of patients who are more likely to benefit from ICIs [124,125]. A methodology named "Immunoscore" has been defined to measure intra- and peritumoral T-cell infiltrate [126]. It has a validated prognostic value and provides a reliable estimate of the recurrence risk in colon cancer [127,128]. An immunoscore for NSCLC patients is currently under development [129].

### 3.2. Liquid Biopsy and Emerging Blood-Based Biomarkers

Tissue biopsy is the standard method for cancer diagnosis, histological classification, and molecular diagnostics. However, invasiveness limits its feasibility and repeatability, and tumor lesions are sometimes difficult to biopsy. Furthermore, it can fail to monitor disease evolution and capture tumor heterogeneity. To overcome these limitations, liquid biopsy has become an attractive opportunity for cancer diagnostics. It is a non-invasive tool and consists of analyzing cancer-associated elements in biological fluids, such as blood, urine and saliva. Primarily, blood-based liquid biopsy is one of the main fields in ICI biomarker research. It allows the isolation and analysis of multiple tumor-derived or tumor-associated components circulating in the bloodstream, including circulating tumor cells (CTCs), immune cells, extracellular vesicles (EVs), cell-free DNA (cfDNA) and microRNA (miRNA) [130].

Peripheral immune cell populations could reflect the interaction between the immune system and cancer. Peripheral blood mononuclear cells (PBMC), absolute lymphocyte count, absolute neutrophil count (ANC), absolute eosinophil count, absolute monocytic count, platelet-to-lymphocyte ratio (PLR), lymphocyte-to-monocyte ratio, and neutrophil-to-lymphocyte ratio (NLR) have all been investigated as predictors of response to immunotherapy in different solid cancers [131]. In particular, several studies have reported that a high NLR has a negative predictive and prognostic value. Two recent metanalyses found that elevated blood NLR was associated with poor PFS and OS in NSCLC patients treated with PD-1/PD-L1 antibodies [132,133]. Researchers also explored the baseline and dynamic changes in T lymphocytes, especially PD1 + CD 8 T cells, as a straightforward marker for distinguishing responders and non-responders [134,135].

Recently, a group of immature cells produced by emergency myelopoiesis has been described as a potential biomarker for ICIs. So-called myeloid-derived suppressor cells (MDSCs) are found both in peripheral blood and tumor sites and exert immunosuppressive activity. In particular, they contribute to impair CD8+ T cell and natural killer (NK) cells via secreting cytokines, expressing inhibitory surface markers, or through the upregulation of ligands for co-stimulation, including PD-L1 and CD86 [136,137]. Peripheral MDSC have been proposed as negative predictors of response and prognostic factors in several solid cancers, but there is controversy in studies on lung cancer [138,139]. Interestingly, in a cohort of 61 mNSCLC patients, high levels of circulating monocytic-MDSC showed a negative impact on anti-PD-1 efficacy [140]. On the contrary, in a study on 53 patients, higher baseline levels of granulocytic-MDSC were associated with better response to nivolumab treatment [141]. At present, limited consensus has been reached on NSCLC, and further studies are needed to clarify the potential role of MDSCs and its subsets as immune-related biomarkers.

CTC and cfDNA have been used for assessing PD-L1 expression, TMB and other immune-related signatures. Lower expression of soluble PD-L1 and high blood-TMB appear to correlate with good response to ICIs [142,143]. However, the main evidence comes from retrospective exploratory analysis or small prospective series [144,145]. Additionally, the studies evaluating concordance between tissue and liquid-biopsy have produced conflicting results [146]. Moreover, multiple detection diagnostics have been used, as workflow and protocols are not harmonized [147].

In general, despite the potential advantages, due to the lack of standardized methodologies, the low accuracy rates and controversial findings, blood-based biomarkers are not yet validated in the clinical setting.

### 3.3. miRNA

Among novel blood-biomarkers, microRNAs are attracting great interest. TME hosts many nanosized particles, named extracellular vesicles (EV), which are produced and released into the bloodstream by various cell types, including cancer cells and immune cells [148]. EVs represent a heterogeneous group of membranous structures, classified according to cellular origin and biological functions. Two subtypes, exosomes and plasma

membrane-derived microparticles (microvesicles), are primarily considered mediators of intercellular communication, and there is evidence of their involvement in tumor growth, metastasis and immunomodulation [149,150]. EVs-derived microRNAs, a single-stranded non-coding RNA, are believed to be the main regulators of these processes. They act at the post-transcriptional and translational level for various cellular functions. More and more evidence indicates a crucial role in the development, maturation and activation of immune cells [151–154]. Therefore, miRNAs have emerged as key players in tumor immunity and represent potential biomarkers for immunotherapy.

Recent studies have explored the role of plasma-derived miRNA signatures in NSCLC patients treated with ICIs. In a cohort of patients (n = 80) treated with Nivolumab, a 10-high expressed miRNA pattern was associated with statistically significant improvement in PFS and OS [155]. A plasma exosomal miRNA profile (Hsa-miR-320d, Hsa-miR-320c and Hsa-miR-320b) was identified as a potential predictor of efficacy for anti-PD-1 treatment [156]. In a consecutive series of 140 patients receiving ICI treatment, a plasma immune-related miRNA-signature classifier (MSC) composed of 24 miRNAs was correlated with ORR, PFS and OS. Significance was greater if combining MSC with PD-L1 expression. The authors concluded that MSC could supplement PD-L1 to identify NSCLC patients with worse response and survival [157]. Moreover, Cortez et al. found that PD-L1 was regulated by p53 via miR-34, which directly binds to the PDL1 3′ untranslated region in models of NSCLC [158]. In a cohort of 88 NSCLC patients treated with anti-PD-1 antibodies, the EV-miR-625-5p level was found to act as an independent biomarker of response and survival in ICI-treated NSCLC patients, in particular in patients with PD-L1 expression $\geq$ 50% [159].

Undoubtedly, further studies are needed to clarify the role of microRNAs and identify which signatures can better predict responses to ICI. Detection methods also need to be refined and standardized. However, the preliminary data available in the literature are promising and pose a new challenge for clinical and translational research: microRNAs as novel biomarkers for NSCLC immunotherapy.

### 3.4. Microbiota

While lungs were previously considered sterile organs, as bacteria were rarely isolated from healthy lungs using conventional culture techniques, it is now known that healthy lungs host a complex community of microbes, collectively named microbiota, which oversees maintaining homeostasis of the local environment and lung wellness overall.

In healthy lungs, microbial density is low, significantly lower when compared with gut microbiota, counting 103 cells/g to 105 cells/g of tissue, and it is determined by microbial immigration from the upper respiratory tract through microaspiration of pharyngeal secretion, a passive process involving the oral and pharyngeal muscles, and microbial elimination, via mucociliary clearance, cough and host immune defense [160]. Qualitatively, healthy lungs normally contain a high bacterial diversity and distinct variability at the species level, made up of the phyla Bacteroidetes and Firmicutes, Prevotella, Veillonella, and Streptococcus [161,162]. In patients with lung diseases, microbiome dysbiosis is a common finding. In these cases, the balance between microbial immigration and elimination seems to be disrupted; hence, the lung microbiota becomes altered, leading to a decrease in community diversity and an increase in pathogenic bacteria. This may result in the activation of inflammatory pathways with the upregulation of inflammatory mediators and cytokines, which trigger epithelial cell proliferation, thus promoting the development of tumors. It was shown that in patients with lung cancer, there was a significant enrichment of Granulicatella, Abiotrophia, and Streptococcus and decreased community diversity [163]. In 2015, Boursi et al. carried out a large cohort study demonstrating that recurrent exposure to certain antibiotics (penicillin, cephalosporins or macrolides) resulted in the increased risk to develop lung cancer, as they could alter bacterial abundance, composition, and diversity, thus inducing dysbiosis [164].

Recently, many studies have focused on the "gut-lung axis", since the lung and gut microbiomes influence each other through a complex bidirectional lymphatic and blood

communication [165]. For instance, it has been shown that high levels of Enterococcus in gut microbiota are associated with lung cancer [166]. It was postulated that the composition of gut microbiota might influence the cancer immune response and efficacy of immunotherapy [167]. Routy et al. reported a positive correlation between abundance of Akkermansia muciniphila species in the gut microbiota with a higher response to anti-PD1 therapy in lung cancer patients. Additionally, patients treated with antibiotics early before, during, or shortly after treatment with immunotherapy, presented a significantly decreased efficiency against lung cancer [168]. Furthermore, prolonged PFS to anti-PD-1 therapy was observed in NSCLC patients with high microbiome diversity compared with those with low diversity [169].

Nowadays, there is little evidence on the role of the microbiome in lung cancer, although there is a biological rationale and preliminary data on potential predictive biomarkers for immunotherapy. Consequently, analysis of the patient's microbiome through bronchoalveolar lavage, saliva or stool samples is not yet routinely used in clinical practice but should be encouraged in a research setting.

### 3.5. Mechanisms of Resistance

NSCLCs are among the most responsive cancers to ICIs, but there is a consistent proportion of patients who do not benefit from treatment and are considered "resistant". Resistance can be defined as "primary", when there is a lack of response, or "acquired", when patients initially respond to immunotherapy and then develop progression of disease. These mechanisms may be intrinsically related to the tumor cells or may be caused by extrinsic factors, which include host-related factors and the tumor microenvironment (TME) [170]. Firstly, oncogenic drivers, which define the oncogene-addicted disease, represent one of the main causes of primary resistance to immunotherapy. EGFR- and ALK-positive lung cancers have a scarce release of antigens as well as show a reduced expression of T CD8+ cells in their tumor infiltrate, which results in a reduced response to ICIs. Similarly, tumors with negative expression of PD-L1 or low TMB are more easily characterized by low antigenic charge, which hardly elicits a significant immune response [171,172]. Secondly, conditions intrinsic to the TME, including multiple immune cell populations infiltrating the tumor, tumor cells and stromal cells, can be predictive of the host response to immunotherapy. The composition of the TME can be described based on the pattern of the immune infiltration. An "immune-inflamed" tumor is characterized by the infiltration of CD8+ and CD4+ T cells, B cells and NK cells, indicating a better response to ICI. In contrast, "immune-desert" phenotype is characterized by the absence of abundant T cells, thus indicating a poor immune response [173]. Moreover, the phenotype of tumor-infiltrating lymphocytes (TILs) can also influence the efficacy of immunotherapy. There seems to be a correlation between TIL clonality and the response to immunotherapy, considering that immune cell response requires the activation and expansion of specific antigen-reactive T-cell clones [174]. Recently, STK11 and KEAP1 mutations were associated to KRAS status and described as predictors of resistance to ICI. However, studies showed similar outcomes among patients treated with ICI or other treatments, suggesting they are prognostic, and not predictive, factors in NSCLC [175].

Strategies to overcome therapeutic resistance to immunotherapy and improve the activity of these treatments require multiple efforts, both to convert "cold tumors" into "hot tumors" and enhance the host immune cells through triggering T cells. It seems that chemotherapy also holds immunostimulant activity, mostly due to the release of immunogenic molecules from dying tumor cells. Furthermore, ionizing radiations are also known to elicit immune activation through the release of damage-associated molecular patter molecules (DAMPs), recognized by the immune system, thus potentially enhancing the immune response of the host. Immunotherapeutic combination strategies, acting on broader pathways of the immune response, could also overcome some mechanisms of resistance to single agents [176,177]. However, the mechanisms of tumor immune resistance are very complex and not really understood, remaining an area of clinical challenge [178].

The in-depth exploration of immune resistance mechanisms may contribute to the discovery of new therapeutic targets, predict treatment efficacy and adapt the treatment regimen in a timely manner.

## 4. Conclusions

Immune checkpoint inhibitors have revolutionized the treatment paradigm of advanced non-oncogene addicted NSCLC and significantly improved patients' prognosis. However, the initial enthusiasm was later dwarfed when it was discovered that only a minority of patients have lasting responses. Despite combination strategies and novel agents further increasing response rates, it remains unclear why most patients develop progressive disease. PD-L1 expression remains the most widely used biomarker in clinical practice, but alone it cannot accurately predict durable response and survival, nor can it reflect the complex mechanisms that occur in TME during ICI treatment. Further studies should focus on identifying biomarkers for prediction treatment responses and resistances to optimize patient selection and guide therapeutic choices.

**Author Contributions:** Conceptualization, M.R.; methodology, M.R. and G.L.C.; writing—original draft preparation, M.R. and G.L.C.; writing—review and editing, F.C., A.G., A.C., A.V., G.R.D.F., I.F., F.P., L.B. and V.S.; supervision, A.C., G.T. and B.V. All authors have read and agreed to the published version of the manuscript.

**Funding:** This research received no external funding.

**Conflicts of Interest:** The authors declare no conflict of interest.

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
