# Peer review of "Immunotherapy for Metastatic Non-Small Cell Lung Cancer: Therapeutic Advances and Biomarkers"

_curroncol, doi:10.3390/curroncol30020181_

Round 1

Reviewer 1 Report (Previous Reviewer 2)

The manuscript has improved a lot. It would be better if the conclusion could add one sentence summarizing the biomarkers mentioned in the manuscript (for example the ones shown in Fig 1), since "biomarker" is in the title of this review article. Furthermore, small grammer mistakes and typos are detected.

Author Response

Dear Reviewer, thank you very much for your comments.

We followed your advice and changed the conclusion as follows:

“…PD-L1 expression remains the most widely used biomarker in clinical practice, but alone it cannot accurately predicts durable response and survival, nor can it reflects the complex mechanisms that occur in TME during ICI treatment. More prospective trials in NSCLC patients could clarify the role of TMB and TILS, and of emerging blood-based biomarkers such as peripheral immune cells phenotypes and miRNA signatures. Further studies should focus on both the prediction of treatment responses and resistances, in order to optimize patient selection and guide therapeutic choices.”

Reviewer 2 Report (Previous Reviewer 1)

Minor comment: 

As for your reply to my comment "We changed the statement related to survival: “...Although it is still difficult to accurately estimate the overall survival benefit, some studies reported increased survival rates 5 times higher than those achieved with chemotherapy [8,9]."

I believe is possible as well as easy (with appropriate data source) to accurately estimate the overall survival benefit of patients with advanced NSCLC. However it is not clear if the marketing of immunotherapy really improved the survival gain of patients with advanced NSCLC (PMID: 34885238; PMID: 32470017). Please change this sentence since there is still contrasting evidence 

Author Response

Dear Reviewer,

  we modified the text and added two references following your suggestions.

“… Several studies reported significant survival benefits over chemotherapy, reaching median survivals between 11 and 26 months [8]. Nonetheless, it is not clear if the survival gain shown in clinical trials is translatable to a real world patient population [9]”

[8] Zhang, X.; Xu, Q.; Yu, X.; et al. What Is Long-Term Survival and Which First-Line Immunotherapy Brings Long-Term Survival for Advanced Wild-Type Non-Small Cell Lung Cancer: A Network Meta-Analysis Based on Integrated Analysis. Front Immunol. 2022 Apr 5;13:764643. doi: 10.3389/fimmu.2022.764643. PMID: 35450068; PMCID: PMC9016897.

[9] Wallrabenstein, T.; Del Rio, J.; Templeton, AJ.; Buess, M. Much has changed in the last decade except overall survival: A Swiss single center analysis of treatment and survival in patients with stage IV non-small cell lung cancer. PLoS One. 2020 May 29;15(5):e0233768. doi: 10.1371/journal.pone.0233768. PMID: 32470017; PMCID: PMC7259780.

Thanks again for your comments

This manuscript is a resubmission of an earlier submission. The following is a list of the peer review reports and author responses from that submission.

Round 1

Reviewer 1 Report

Dear authors, your study reported an overview of the immunotherapy treatment with a look to therapeutic advances and biomarkers. I have some comments

- In the present literature, a lot of reviews have been published on the advances of immunotherapy in the last years (only few examples: 32368178, PMID: 33979196, PMID: 32450594). Please explain the novelty of your work in relation to other works. I suggest authors to focus more deeply on the innovative aspects.

- In your article you did not mention (or partially) the potential association of immunotherapy to anti-angiogenic drugs (PMID: 32913278). Please report it.

Other comments:

Line 48-50: You stated that “The use of the Immune Checkpoints Inhibitors (ICIs) has revolutionized the management of advanced NSCLC and improved patients’ prognosis, achieving survival rates 5 times higher than those achieved with chemotherapy.” I recommend the authors to be still cautious about the statement related to survival. There are several studies conducted in real-word setting that still report low survival increase for patients with advanced NSCLC over the last decade (e.g.  PMID: 34885238; PMID: 32470017) even after ICIs introduction. Please introduce better this aspect reporting there is still controversial evidence on this matter.

Line 65-76: I suggest authors to avoid a deep explanation of mechanism of actions of CTLA-4 and PD-1/PD-L1 since there is a lot of literature on it. Given the title of your review “ therapeutic advances and biomarkers “ maybe authors could focus more on novel immunotherapies (line 83-86).

Line 99-105: Immunotherapy is related also to an hyper progression of the disease in some patients. Please report it here.

Line 111 please report citations.

Line 256: You already stated it in the previous chapters, please avoid reporting.

Author Response

- In the present literature, a lot of reviews have been published on the advances of immunotherapy in the last years (only few examples: 32368178, PMID: 33979196, PMID: 32450594). Please explain the novelty of your work in relation to other works. I suggest authors to focus more deeply on the innovative aspects.

Dear Reviewer,

We sincerely thank you for your comments. We revised the paper focusing more on the innovative aspects. We have partially rewritten the abstract (lines 18-19). We added discussion on novel immune checkpoints (lines 77-101). We implemented the section on main clinical challenges by also discussing pseudoprogression and hyperprogression, as suggested. Furthermore, we added MDSC as potential novel biomarkers (lines 502-515) and consequently we modified the figure 1. Finally, we rewrote the conclusions. 

- In your article you did not mention (or partially) the potential association of immunotherapy to anti-angiogenic drugs (PMID: 32913278). Please report it.

We reported the potential association of immunotherapy to angiogenic drugs (lines 265-278) and added references [64-67]

Other comments:

Line 48-50: You stated that “The use of the Immune Checkpoints Inhibitors (ICIs) has revolutionized the management of advanced NSCLC and improved patients’ prognosis, achieving survival rates 5 times higher than those achieved with chemotherapy.” I recommend the authors to be still cautious about the statement related to survival. There are several studies conducted in real-word setting that still report low survival increase for patients with advanced NSCLC over the last decade (e.g.  PMID: 34885238; PMID: 32470017) even after ICIs introduction. Please introduce better this aspect reporting there is still controversial evidence on this matter.

We changed the statement related to survival: “…Although it is still difficult to accurately estimate the overall survival benefit, some studies reported increased survival rates 5 times higher than those achieved with chemotherapy [8,9]. “ (lines 49-51)

Line 65-76: I suggest authors to avoid a deep explanation of mechanism of actions of CTLA-4 and PD-1/PD-L1 since there is a lot of literature on it. Given the title of your review “ therapeutic advances and biomarkers “ maybe authors could focus more on novel immunotherapies (line 83-86).

Thanks, we've removed deep explanation of mechanism of actions of CTLA-4 and PD-1/PD-L1 antibodies. We added discussion on novel immune checkpoints (lines 77-101).

Line 99-105: Immunotherapy is related also to an hyper progression of the disease in some patients. Please report it here.

We've boosted the discussion on pseudoprogression and hyperprogression (340-365).

Line 111 please report citations.

Done

Line 256: You already stated it in the previous chapters, please avoid reporting. 

Done

We thank you for the suggestions. We believe that the changes have improved the paper, hoping it is now more suitable for publication.

Dr. Marco Russano

Campus Bio-Medico University of Rome, Via Alvaro del Portillo, 21 - 00128 Rome, Italy

Reviewer 2 Report

The authors would like to review the updates on immunotherapy and novel biomarkers in NSCLC. However, the main weakness of this review is that many references are missing, the information is not uptodate/comprehensive enough, and the sections are not well-organized or too short:

1. References should be added for the first statments in the introduction. In the second paragraph, " a subset of..." need to be clarified

2. "until a few years ago, chemotheray was the only..."--> may not be correct, EGFR inbitor has been introduced for about a decade? And should add the discussion about the comparison between adding IO vs adding TKI

3. line 83 "new immune checkpoint targets" need to describe more, include at least 3 references

4. line 100 should clarify what are the "combination therapies" 

5. line 215 It is not clear why the authors suggest there is still a favorable trend while there was no statistical significance

6. line 290 talk about the combination therapy of IO with TKI, need to elaborate into another section

7. line 28 about NGS for identification for IO need to elaborate into a new section, and tumor mutation burden should be added

8. line 343 need to elaborate the current tissue-based immune biomarkers

9. section 3.2 and figure 1 should include the data on MDSC

Author Response

The authors would like to review the updates on immunotherapy and novel biomarkers in NSCLC. However, the main weakness of this review is that many references are missing, the information is not uptodate/comprehensive enough, and the sections are not well-organized or too short:

Dear Reviewer, We sincerely appreciate your honesty thank you for your comments. We have added 25 references compared to the previous version. We have revised the paper focusing more on the updates and innovative aspect. We added discussion on novel immune checkpoints (lines 77-101). We implemented the section on main clinical challenges by also discussing pseudoprogression and hyperprogression.  Furthermore, we added MDSC as potential novel biomarkers (lines 502-515) and consequently we modified the figure 1. Finally, we rewrote the conclusions. 

  1. References should be added for the first statments in the introduction. In the second paragraph, " a subset of..." need to be clarified

Done (introduction section).

  1. "until a few years ago, chemotheray was the only..."--> may not be correct, EGFR inbitor has been introduced for about a decade? And should add the discussion about the comparison between adding IO vs adding TKI

We decided to remove the statement. We rewrite the final part of introduction (lines 46-55)

  1. line 83 "new immune checkpoint targets" need to describe more, include at least 3 references

We added discussion on novel immune checkpoints (lines 77-101) (references 20-28).

  1. line 100 should clarify what are the "combination therapies" 

Done (lines 376-377)

  1. line 215 It is not clear why the authors suggest there is still a favorable trend while there was no statistical significance

Statement not necessary. We have removed it.

  1. line 290 talk about the combination therapy of IO with TKI, need to elaborate into another section

In our opinion, adding a new section is superfluous and not in line with the subject of this review. The TKI-IO associations should be discussed in a review on “oncogene-addicted disease”. In contrast, we have focused on non-oncogene addicted NSCLC. However, we have discussed it in section 2.3

  1. line 28 about NGS for identification for IO need to elaborate into a new section, and tumor mutation burden should be added

In the revised manuscript, we discuss the role of the NGS (lines 305-317) and TMB (435-456)

  1. line 343 need to elaborate the current tissue-based immune biomarkers

We have dedicated section 3.1 to tissue-based immune biomarkers.

  1. section 3.2 and figure 1 should include the data on MDSC

We discussed MDSC as potential novel biomarkers (lines 502-515) and consequently we modified the figure 1. Thank you so much for this suggestion.

We thank you for your comments. We believe that the changes have improved the paper, hoping it is now more suitable for publication.

Dr. Marco Russano

Campus Bio-Medico University of Rome, Via Alvaro del Portillo, 21 - 00128 Rome, Italy
